# Rapid and high-resolution patterning of microstructure and composition in organic semiconductors using 'molecular gates'

Aleksandr Perevedentsev [1✉] & Mariano Campoy-Quiles [1✉]

Photolithography has been a major enabling tool for miniaturisation of silicon devices that underpinned the electronics revolution. Rapid, high-resolution patterning of key material characteristics would, similarly, accelerate the advent of molecular electronics and photonics. Here we advance a versatile approach employing local diffusion of functional small-molecular compounds through a solution-processed 'molecular gate' interlayer. Diffusion is activated using laser light or solvent vapour jets——a process that can be finely modulated down to molecule-on-demand deposition precision with almost photolithographic resolution (<5 μm) and speeds (3 mm s$^{-1}$). Examples of principal pattern types are presented including molecular conformation for integrated photonics; chain orientation for polarised security features and micro-engineered electronics; and doping with local conductivity values >3 S cm$^{-1}$ for improved electronic devices. Finally, we demonstrate the unique capability for one-step patterning of multiple functionalities by spatially modulating composition in ternary blends, leading to locally tunable photoluminescence from blue to red.

---

[1] Institut de Ciència de Materials de Barcelona (ICMAB-CSIC), Bellaterra 08193, Spain. ✉email: aperevedentsev@icmab.es; mcampoy@icmab.es

We hold the fact to be self-evident that the field of molecular electronics and photonics[1–3] is reaching its promise, with several classes of high-performance, lightweight and flexible devices appearing on the market or nearing commercialisation. Besides potentially novel functionalities that may complement, or go beyond, those of inorganic semiconductors, two of the principal motivations underlying the growth of this field are, arguably, economic and environmental, given the cost and energy efficiency improvements that can be achieved by large-area low-temperature solution-based fabrication.

A key part of fabrication of organic semiconductor active layers involves spatial patterning of material characteristics to enable device-specific functionalities[4–6]. Of these, the principal characteristic is material composition, as required for fabrication of, e.g., interconnects and electrodes for organic field-effect transistors by local doping[7] and emissive components for organic light-emitting diodes (OLEDs)[4]. For polymeric semiconductors, two additional microstructural characteristics are of practical importance, namely (i) chain conformation, which affects refractive index[8] and other optoelectronic properties[9,10], and (ii) chain orientation, which enables anisotropic electronic[11,12] and thermal[13] conductivities, as well as polarised absorption and emission of light[14,15].

Numerous approaches towards patterning organic semiconductors have been reported; excellent reviews can be found in refs. [4,6]. These can be broadly classified into light-based methods such as photolithography[16] and laser-induced forward transfer (LIFT)[17,18]; printing methods such as inkjet, vapour jet and aerosol jet[19–23], and contact methods such as hard and soft imprint lithography[24–26]. Of these, photolithography represents the most mature technology, with a further advantage being its high spatial resolution (~1 µm in large-field proximity-mode implementations). Inkjet printing, on the other hand, offers superior versatility and speed, albeit with lower resolution (typically ~30–50 µm).

The availability of this plurality of patterning techniques is promisingly moving the field forwards, with an increasing number of reports on devices with integrated components. Nevertheless, particularly in the context of optimal large-area, roll-to-roll-type processing, the aforementioned techniques are subject to a number of limitations. First, the available techniques typically are additive or subtractive, requiring an additional back-filling step to return to a planar film format. Second, many of the patterning strategies necessitate slow, multi-step processing as in the case of photolithographic approaches. Third, several structural features remain generally unattainable, notably local patterning of molecular orientation, which is either limited to specific compounds[27], demands the restrictive use of insulating alignment layers[28] or requires very specific geometries[29,30]. Fourth, most patterning approaches are 'binary', with a single structural feature defined on the baseline layer in one patterning step, thus requiring additional time-consuming post-processing or mask alignment in cases when multiple patterns or functionalities are desired.

Recognising these limitations of the state-of-the-art, we advance a conceptually different approach to micro-patterning of organic semiconductors based on donor diffusion through a 'molecular-gate' interlayer, which provides both the spatial resolution of photolithography and the versatility of printing techniques. By selection of appropriate donor compounds, we demonstrate spatial patterning of chain conformation and orientation, as well as material composition, in several benchmark semiconducting polymers. Given the use of vacuum-free solution-based processing and non-contact stimuli such as laser light, the reported patterning approach is expected to be highly suited for rapid micro-patterning of advanced device structures in both roll-to-roll and laboratory-scale environments.

## Results

**The 'molecular-gate' concept.** The concept underlying the patterning method advanced herein is the controlled, stimulus-induced diffusion of functional small molecules into the target film through a permeation-switchable membrane. As illustrated below and in Supplementary Fig. 1, the method employs the following sequential processing steps (the terminology will be adopted hereafter): first, solution-based deposition of a target semiconductor film, a 'molecular-gate' interlayer and a donor layer, which comprises functional small-molecular compound(s). This is followed by application of thermal-, solvent-vapour- or laser-based stimulus to activate diffusion of the functional compound(s) of the donor layer into the target film through the molecular gate. The residual donor layer and molecular gate are subsequently removed in the final processing step. Supplementary Movie 1 shows an animated illustration of the molecular-gate concept and exemplary applications thereof.

The molecular-gate functions as a semipermeable membrane, preventing uncontrolled diffusion of functional compounds into the target film (e.g., during solution-based deposition of donor layer) and allowing diffusion under an external stimulus such as heat or solvent vapour. Using diffusion through a ~100 nm interlayer as the physical mechanism for patterning——rather than droplet transfer as in inkjet printing or LIFT concepts[17]——enables essentially a molecule-by-molecule deposition, similar to the thermal evaporation process but with high spatial resolution, mask-free and no vacuum requirements.

The desired requirements for the molecular-gate material include preferential solubility in solvents orthogonal to those used for depositing typical organic semiconductors and donor compounds, high molecular weight to ensure good film-forming ability, and high thermal stability for minimal inter-diffusion with adjacent layers during patterning. Hence, poly (sodium 4-styrenesulfonate) (pNaSS) was selected, given that it features glass transition and decomposition temperatures of 211 °C and >470 °C, respectively[31,32], with high-molecular-weight materials commercially available at a moderate cost. Its solubility in water and insolubility in common organic solvents allow for its straightforward deposition onto organic semiconductor films and post-patterning removal as well.

Drawing inspiration from our previous work[8,33], here a macroscopic proof-of-concept demonstration of molecular-gate-based patterning is provided using chain-conformation-mediated photoluminescence (PL) switching in poly(9,9-dioctylfluorene) (PFO) films. PFO exhibits a well-known conformational isomer, termed the 'β-phase', corresponding to an extended, planar-zigzag geometry, which thus features a distinctly red-shifted PL spectrum relative to an in-plane-isotropic, glassy sample (PL peaks at ~438 and 423 nm, respectively)[9]. β-Phase formation is induced in glassy PFO by exposure to various small-molecular 'solvents', proceeding via a co-crystallisation process[34,35]. The overall PL is extremely sensitive to the presence of even a small, ≤1% fraction of β-phase chain segments[9], offering an ideal material system for demonstrating the robustness of the proposed patterning approach, as well as the molecule-on-demand concept.

Figure 1 illustrates the individual processing steps. The baseline glassy PFO film is spin-coated on a glass substrate (step 1), exhibiting dark-blue PL under ultraviolet (UV) illumination. Subsequent spin-coating of pNaSS from solutions in water (step 2) yields a homogeneous film without altering the PL characteristics of PFO. The presence of the pNaSS molecular gate protects the PFO film during the following spin-coating of lauric acid (LA)

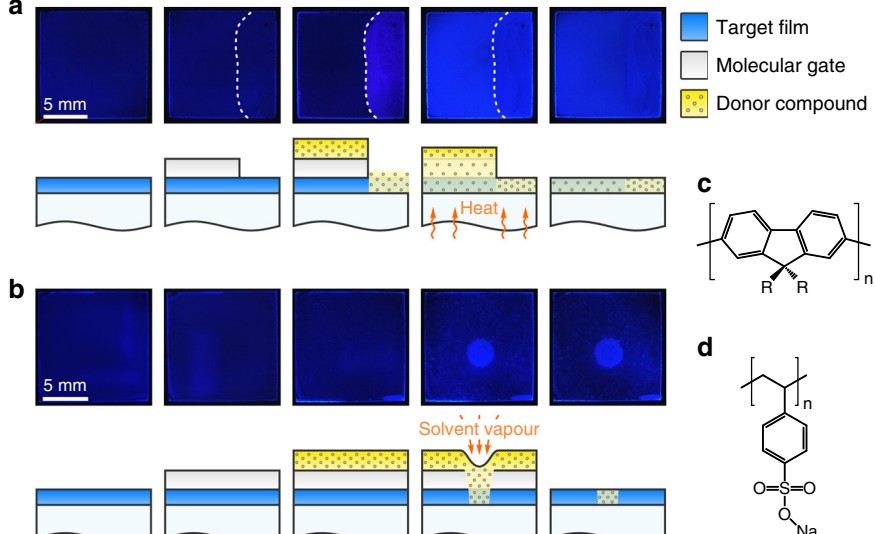

**Fig. 1 Illustration of the molecular-gate concept by PL switching in PFO films.** Examples are shown for (**a**) thermal and (**b**) solvent-vapour-based stimuli. Top rows show photographs of the films under UV light illumination at the sequential processing steps, depicted schematically in the bottom rows. These are: deposition from solutions of the target film (PFO; depicted in blue), the 'molecular gate' (pNaSS; depicted in grey) and the functional donor compound (lauric acid; depicted in yellow + grey circles); followed by application of heat or solvent vapour to diffuse lauric acid into the PFO film via the molecular gate and, finally, removal of the residual donor layer and gate by spin-off. In the second step of (**a**), a part of the gate was removed by immersion in water (as shown by the dotted line) to highlight its role in preserving the 'baseline' characteristics of the target film during donor layer deposition. Also shown are the chemical structures of (**c**) PFO ($R = C_8H_{17}$) and (**d**) pNaSS.

(step 3)—the functional 'solid solvent' employed for inducing the conformational transformation. It is noteworthy that depositing LA directly onto PFO (step 3 in Fig. 1a) immediately induces the β-phase, as evidenced by the appearance of light-blue PL, thereby preventing any opportunities for local patterning.

Controlled diffusion of LA (step 4) into the PFO film is activated by heating slightly above the melting temperature of LA ($T_m = 44$ °C; Fig. 1a) or exposure to solvent vapour (Fig. 1b) that renders the small-molecule mobile, resulting in formation of the β-phase and the corresponding change in PL. Exposure to solvent vapour through a nozzle[33] (step 4 in Fig. 1b) enables a local change in the chain conformation through the interaction of LA with the polymer (note that acetone and water are non-solvents of PFO and, in the absence of LA, do not induce β-phase formation). The residual LA donor layer and pNaSS gate are finally removed by spin-off (step 5) with solvents such as acetone and water (orthogonal to PFO and the majority of other organic semiconductors) without dissolving, or further modifying, the PFO film. Reassuringly, the PL emission colour is found to be very homogeneous for the sample area where LA diffused through the gate—and less so for the area where LA was deposited onto PFO directly (cf. left and right sides of panel (5) in Fig. 1a)—indicating that gated diffusion is a highly controllable process.

Key to realisation of patterning, therefore, are the selection of small-molecular compounds with the desired functionality, as well as delivering a local stimulus for 'opening' the gate and enabling spatially selective diffusion. The stimulus, whether heat- or solvent-vapour-based, is employed to impart molecular mobility to the compounds comprising the donor layer by melting or swelling. The resulting rate of diffusion of said compounds into the donor layer scales exponentially with temperature and is governed by additional parameters such as glass transition of the target semiconductor film, molar mass of the functional compounds and thickness of the molecular gate[36]. As laser light is the most versatile choice of stimulus, the following sections will present local patterning of various feature types by laser-induced 'gated' diffusion.

**Patterning chain conformation.** Laser patterning of the β-phase chain conformation in PFO films is performed in the configuration shown in Fig. 2a, with a more detailed, step-by-step schematic illustration presented in Supplementary Fig. 1. PFO/pNaSS/LA trilayer samples (as above) are deposited on indium tin oxide (ITO)/glass substrates. Laser excitation at 785 nm (non-resonant for PFO) is used to generate local heating at the substrate level by partial absorption within ITO, activating diffusion of LA into PFO in an area that is essentially defined by the excitation spot and laser parameters.

PL microscopy images of β-phase lines patterned by continuous scanning of the sample in the laser focal plane at a constant writing speed $v$ and varying power $P$ are shown in Fig. 2b, with spectral filtering at 438 nm used to preferentially select the peak of β-phase emission[8]. (Full PL spectra are shown in Supplementary Fig. 2a.) As is apparent from β-phase line widths increasing with $P$, laser power represents a key parameter that modulates pattern features by governing the local temperature rise and, therewith, the diffusivity of the functional small molecule. Writing speed is another parameter that determines the time over which the diffusion occurs.

To demonstrate the impact of these parameters, pattern dimensions (full-width at half-maximum (FWHM) of β-phase PL profiles) and contrast (maximum induced β-phase fraction) were extracted as function of $v$ and $P$ (PL profiles are shown in Supplementary Fig. 2b,c). The β-phase fraction, β, was estimated by mapping the Raman intensity ratio, $r_R$, of the 1257 cm$^{-1}$ and 1606 cm$^{-1}$ modes across the line patterns and using the empirical relation ($r_R = 3.4 \times 10^{-3} \beta + 6.0 \times 10^{-2}$) reported between the two variables[9]. An exemplary $r_R$ map is shown in Fig. 2b, with representative Raman spectra given in Supplementary Fig. 2d. The results are presented in Fig. 2c, d and highlight the correlated effects of $P$ and $v$ in the simultaneous tuning of pattern

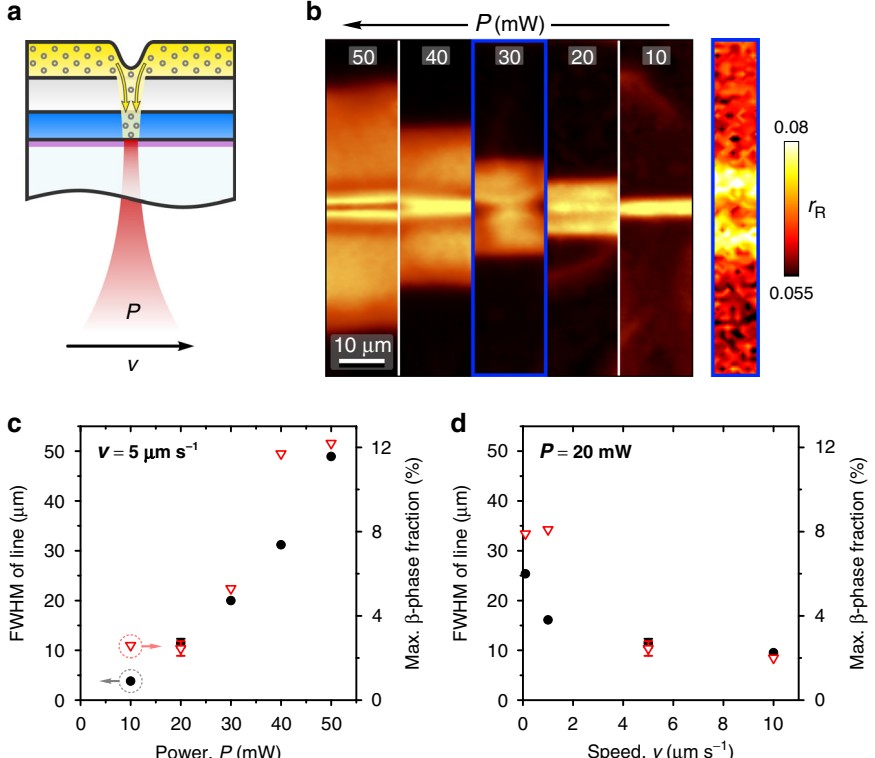

**Fig. 2 Chain-conformation patterning in PFO films. a** Schematic illustration of the laser-patterning process. Colours and symbols as in Fig. 1. ITO layer is shown by a purple stripe; yellow arrows indicate diffusion of donor. **b** Confocal PL microscopy images of β-phase lines patterned in glassy PFO using laser excitation at 785 nm, constant writing speed $v = 5\,\mu m\,s^{-1}$ and laser power $P = 10$–50 mW (as indicated). PL intensity is shown at 438 nm to spectrally select β-phase emission. Also shown in **b** is an exemplary Raman intensity ratio image (right panel) of the line patterned using $P = 30$ mW, recorded as the ratio of Raman intensities, $r_R$, at 1257 and 1606 cm$^{-1}$, from which local β-phase fraction is estimated. **c, d** FWHM of β-phase line patterns (black dot, left ordinates) and maximum induced β-phase fraction in the patterns (red triangle, right ordinates) as function of (**c**) laser power for $v = 5\,\mu m\,s^{-1}$ and (**d**) writing speed for $P = 20$ mW.

dimensions and contrast. For instance, reducing laser power from 50 to 10 mW for a constant $v = 5\,\mu m\,s^{-1}$ (as in Fig. 2b) enables a ten- and sixfold reduction of, respectively, the width and contrast of β-phase patterns.

Due to the increase of refractive index resulting from the glassy-to-β-phase conformational transition[8], the demonstrated β-phase patterning provides a straightforward approach towards the fabrication of photonic structures such as waveguides and refractive index gratings[8,33], while the use of ITO/glass substrates enables natural integration into optoelectronic devices. A pattern resolution of ~4 μm is achieved despite limiting factors such as lateral heat propagation within ITO and dispersive diffusion of LA within the relatively thick (~200 nm) gate interlayer. This resolution is ~50 times better than was obtained previously with vapour printing[33] and only a factor of ~4 larger than demonstrated for dip-pen nanolithography[8]—a technique for which patterning speeds are generally limited to <1 μm s$^{-1}$. Moreover, further improvements can be enabled by addressing the above-mentioned issues, with additional tuning of laser excitation required to prevent overheating-induced 'train-rail' profiles evident for the lines written at high $P$ (Fig. 2b and Supplementary Fig. 2b, c).

Finally, given the polymer : solvent co-crystal structure of β-phase PFO one can estimate the amount of LA that was diffused into the film during patterning. Using the known molar volume of LA and the 'cavity' volume for β-phase PFO[34], a stoichiometry of 1 : 1 is predicted, implying that for the patterns with the lowest achieved β-phase fraction of ~2% the composition of LA molecules to PFO repeat units is ~1 : 50. (In practice, even lower

β-phase fractions can be obtained, although it is noted that quantifying them by spectroscopic Raman mapping becomes increasingly prone to uncertainties.) Considering that full optimisation was not undertaken, such fine control of the diffusion rate clearly exemplifies the aforementioned molecule-on-demand concept.

**Patterning chain orientation.** Besides enabling intra-molecular rearrangement, as in the previous examples, certain crystallisable solvents can also induce directional orientation of semicrystalline polymers by epitaxial solidification[14,15,28]. The principal requirements are the crystal lattice match between the fast growth axis of the small-molecular compound and the $c$-axis of the polymer, and the crystallisation of the small-molecular compound preceding that of the polymer[28]. The first of these can be satisfied by selection of appropriate small molecules, whereas the second is accomplished by employing, for instance, hypoeutectic compositions.

Here we demonstrate laser patterning of directional chain orientation in isotropic films of poly(3-hexylthiophene) (P3HT) —one of the most widely studied semiconducting polymers— using the above-described approach combined with molecular-gate-based processing. 2,1,3-benzothiadiazole (BT) is selected as the crystallisable solvent instead of the more commonly used 1,3,5-trichlorobenzene[14,15,28], due to its comparable periodicity along the fast growth axis ($c_{BT} = 3.85$ Å; $c_{P3HT}/2 \approx 3.8$ Å)[14,37] and a lower melting temperature ($T_m \approx 44$ °C), which facilitates patterning by laser-induced heating. Additional information on

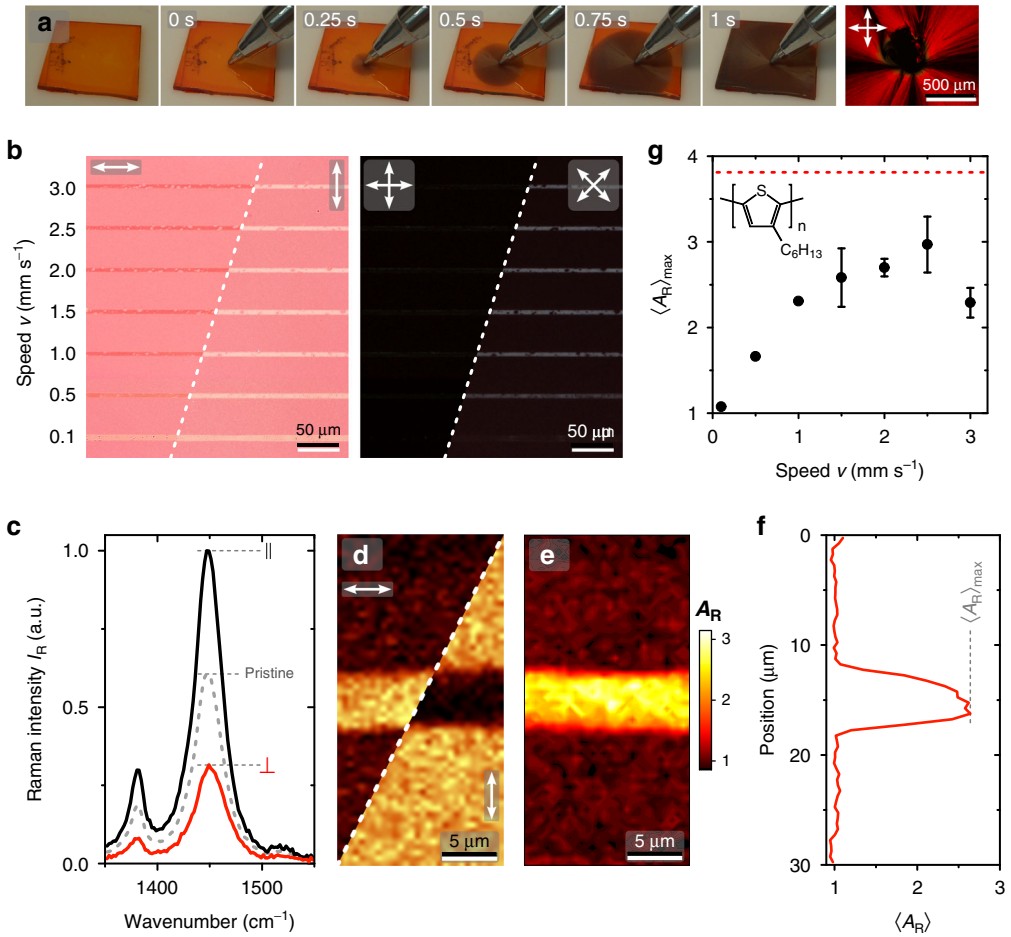

**Fig. 3 Chain orientation patterning in P3HT films. a** Directional crystallisation of a P3HT : BT film by nucleation with BT. Photographs are shown at the indicated times. Also shown is a cross-polarised micrograph recorded at the centre of the obtained P3HT film following sublimation of BT. **b** Transmitted-light micrographs of a P3HT film laser-patterned at the indicated writing speeds $v$. Polarised-incidence and cross-polarised micrographs are shown, with the corresponding relative orientations of polariser transmission axes indicated by the arrows. **c**–**f** Exemplary spatially resolved Raman spectroscopy analysis of the line patterned using $v = 2.5$ mm s$^{-1}$. **c** Polarised Raman spectra centred on the symmetric C=C stretching mode of the P3HT recorded at the centre of the pattern (black and red lines) and pristine unpatterned area (dotted grey line). **d** Polarised Raman maps showing maximum intensity $I_R$ at ~1450 cm$^{-1}$ recorded for the indicated excitation/detection polarisations, and (**e**) the corresponding Raman anisotropy image ($A_R = I_{R,\parallel}/I_{R,\perp}$). **f** Average Raman anisotropy $\langle A_R \rangle$ profile across the pattered line. **g** Maximum values of $\langle A_R \rangle$ as a function of speed $v$. Dotted red line indicates the corresponding value obtained for a macroscopically oriented film, as in **a**. Inset shows the chemical structure of P3HT.

composition-dependent dissolution and re-crystallisation of P3HT : BT blends is given in Supplementary Fig. 3.

A proof-of-principle demonstration of BT-induced orientation of P3HT is shown in Fig. 3a for a freshly spin-coated P3HT : BT blend film (see Supplementary Movie 2). Initially, the film comprises a thin layer of a supercooled solution displaying the characteristic orange colour of dissolved P3HT. Creating a nucleation site—here simply by touching the film with a ball-point pen at time $t = 0$ s—initiates radial crystal growth of BT and, therewith, P3HT at a speed of ~8 mm s$^{-1}$ (note the emergence of dark-red colour of crystalline P3HT). The cross-polarised micrograph recorded at the centre of the film following removal of BT by sublimation confirms radial chain orientation of P3HT (Fig. 3a).

To demonstrate fine spatial control of chain orientation, we again utilise the molecular-gate concept. In the following, the sample geometry for laser patterning using excitation at 785 nm consists of a P3HT film on ITO/glass substrate, capped with a pNaSS film and overlaid on top with a solid BT layer (full details in the 'Methods' section).

Polarised transmitted-light micrographs of patterned P3HT films (Fig. 3b) clearly indicate optical anisotropy within the line patterns obtained at $v \geq 0.5$ mm s$^{-1}$, for which higher absorbance parallel to the long axis indicates preferential orientation of chains, and the corresponding average dipole moments, along the laser scanning direction. Cross-polarised micrographs (Fig. 3b) confirm the presence of birefringence arising from directional crystallisation of P3HT. Additional large-area micrographs highlighting the homogeneity of patterned features are given in Supplementary Fig. 4.

Detailed analysis of molecular anisotropy of P3HT within the patterns is performed using polarised spectroscopic Raman mapping of the 1450 cm$^{-1}$ mode (C=C in-plane symmetric stretching) characteristics. Figure 3c shows exemplary Raman spectra recorded at the centre of a line pattern, revealing higher Raman intensity for polarisation parallel to the writing direction—consistent with the results of optical microscopy. A ~2 cm$^{-1}$ shift of the 1450 cm$^{-1}$ mode to higher energies for the perpendicular-polarised Raman spectrum corroborates the presence of predominantly disordered chain segments[15] perpendicular to the

writing direction. Raman anisotropy, $A_R$, was quantified as the ratio of maximum intensities, $I_R$, at ~1450 cm$^{-1}$ recorded with polarisations parallel/perpendicular to the writing direction. Figure 3d–f illustrate the analysis steps, namely the $I_R$ maps for the two different polarisations (Fig. 3d) and the resulting $A_R$ map (Fig. 3e), from which the profile of average $A_R$ across the pattern is calculated (Fig. 3f).

The maximum value of average Raman anisotropy, $\langle A_R \rangle_{max}$, for the patterns is plotted as a function of $v$ in Fig. 3g, providing further insight into the patterning process. $\langle A_R \rangle_{max}$ peaks at ~3 for $v = 2.5$ mm s$^{-1}$, with higher speeds leading to a roll-off due to an insufficient temperature rise and/or limited diffusion of BT across the gate, whereas lower speeds result in depletion of BT from the illuminated area leading to isotropic crystallisation of P3HT. The highest value of $\langle A_R \rangle_{max}$ obtained within the patterns (Fig. 3g) is somewhat lower than that recorded for a macroscopically oriented film, indicating the possibility of further improvements.

Hence, laser patterning of chain orientation is demonstrated for a model semicrystalline semiconducting polymer, yielding features with FWHM $\approx 4.6 \pm 0.3$ μm written at speeds up to 3 mm s$^{-1}$. Such structures present opportunities for local, micrometre-scale engineering of optical, electronic and thermal-transport properties in thin-film optoelectronic devices, as well as further applications such as security features, of which Fig. 3b presents a first example. We note that the use of the molecular gate is the key enabling feature in the demonstrated process, with its absence—or insufficient thickness—resulting in random patterns of orientation dictated by that of the crystalline BT donor layer (Supplementary Fig. 5). Topography analysis of line patterns of chain orientation by atomic force microscopy (AFM) reveals only a minor ~2% reduction of the total film thickness in the patterned regions resulting from the densification that accompanies polymer crystallisation (Supplementary Fig. 6). This stands in stark contrast to RMS roughness values ~30 nm[14], and in some cases exceeding 100 nm[15], for P3HT films conventionally oriented using solid solvents, highlighting the confinement action[38] provided by the molecular gate, which would facilitate integration of such patterned films into devices.

**Patterning local doping.** Although in the previous examples the functional small molecules were removed after patterning by spin-off or sublimation, here we seek to pattern material composition by retaining the small-molecular component. As a relevant and timely example[39], we use patterning of electrical conductivity in the high-mobility semiconductor poly(2,5-bis(3-tetradecylthiophen-2-yl)thieno-[3,2-b]thiophene) (PBTTT) by p-doping with the Lewis acid tris(pentafluorophenyl)borane (BCF)[40] (chemical structures in Fig. 4a). BCF has been previously reported to be an effective dopant for P3HT[41] and other (macro-) molecular semiconductors[42], but, to our knowledge, has not been used with PBTTT to date.

As in the previous examples, heating a trilayer structure comprising PBTTT, molecular-gate and the small-molecular BCF dopant induces diffusion of BCF across the gate and doping of the semiconducting polymer layer. The conductivity of PBTTT as a function of annealing temperature $T$ follows a sigmoidal evolution (Fig. 4b; measured following spin-off of the auxiliary layers) and reaches a maximum value of 62 S cm$^{-1}$, surpassing ~4 S cm$^{-1}$ obtained for the similarly straightforward solution-based doping using the more common molecular acceptor 2,3,5,6-tetrafluoro-7,7,8,8-tetracyanoquinodimethane (F$_4$TCNQ)[43,44]. (Note that higher conductivities can be achieved for PBTTT:F$_4$TCNQ at a macroscopic level via manipulation of blend microstructure, doping mechanism[45] and molecular orientation[46], as well as the

use of vapour-phase doping[44,45].) Two further features of these results are highlighted for their role in enabling the patterning presented below. First, in the absence of annealing, the PBTTT film retains its low conductivity (Fig. 4b). Second, the spin-off procedure does not appreciably de-dope PBTTT, as illustrated in Supplementary Fig. 7.

The corresponding transmitted-light micrographs (Fig. 4c) evidence a progressive colour change from red to faint-pink upon doping, providing a visual indication of increased electrical conductivity of PBTTT. This change can be understood by reference to the corresponding absorption spectra (Supplementary Fig. 8) showing the emergence of a broad feature centred at ~830 nm (PBTTT cations and BCF anions)[41] and simultaneous attenuation of the peak at ~552 nm ($S_0$–$S_1$ absorption of PBTTT), with an isosbestic point located at 637 nm.

To estimate the conductivity within the patterns (vide infra), an indirect analysis using spectroscopic Raman mapping was employed. Raman spectra of reference macroscopically doped PBTTT films (as in Fig. 4b, c) were recorded over large, ~4 × 4 mm$^2$ sample areas, to minimise uncertainties arising from any inhomogeneities at intermediate $T$ (cf. Fig. 4c) and to provide a better correlation with the large-area conductivity measurements. Selected Raman spectra as a function of conductivity are shown in Fig. 4d. Previous reports have noted the doping-induced increase in the Raman intensity ratio, $r_R$, of the 1393 cm$^{-1}$ mode (thiophene C–C stretch) relative to the 1417 cm$^{-1}$ mode (thienothiophene C=C stretch)[47,48]. Hence, we use $r_R$ plotted against conductivity $\sigma$ (Fig. 4e) as a reference for subsequent analysis, whereby a fit of the data using an empirical equation $r_R = A\sigma^b + C$ ($A = 0.23$, $b = 0.39$ and $C = 0.84$) provides a 'calibration' for $\sigma$ from $r_R$.

Unlike in the previous examples, here patterning via laser-induced heating is enabled by *resonant* excitation at 532 nm—close to the absorption peak of PBTTT (Supplementary Fig. 8). Figure 4f shows that the conductivity obtained within patterned regions as function of laser power $P$ and speed $v$ exhibits the expected dependency, with higher $P$ yielding increased conductivity at highest $v$ due to enhanced temperature rise within the exposure time. The saturation and eventual roll-off for low $v$, on the other hand, are attributed to excessive temperature rise and, therewith, de-doping of PBTTT (see Supplementary Fig. 9 for illustration of the above). A closer look at pattern dimensions as function of $P$ and $v$ provides further insight into the patterning process. As shown in Supplementary Fig. 10b and Supplementary Table 1, reducing writing speed (i.e., increasing the effective diffusion time) by a factor of 600 leads only to a minor factor of 1.6 average increase in pattern dimensions. This indicates that the parasitic in-plane diffusion component is relatively small, as can be expected from an Arrhenius-type exponential dependence of diffusion coefficient on temperature[36], with the latter decreasing sharply outside the laser-illuminated area. On the contrary, increasing laser power by a factor of 2 leads to a factor of 2.2 average increase in pattern dimensions, indicating that the laser-heating-induced temperature rise is the primary parameter governing pattern dimensions.

As a relevant example of potential applications, Fig. 4g shows PBTTT patterned with a radio-frequency identification (RFID)-antenna-type structure, with additional analysis given in Fig. 4h, i. FWHM dimensions of the individual lines are 18 μm, with maximum conductivity reaching ~3 S cm$^{-1}$, whereas the film areas outside the pattern retain the low conductivity of pristine PBTTT (Fig. 4i).

Here we note yet another important enabling feature of the molecular-gate concept, namely that the presence of the hydrophilic pNaSS interlayer, in fact, provides optimal wetting and facilitates deposition of homogeneous BCF donor layers from

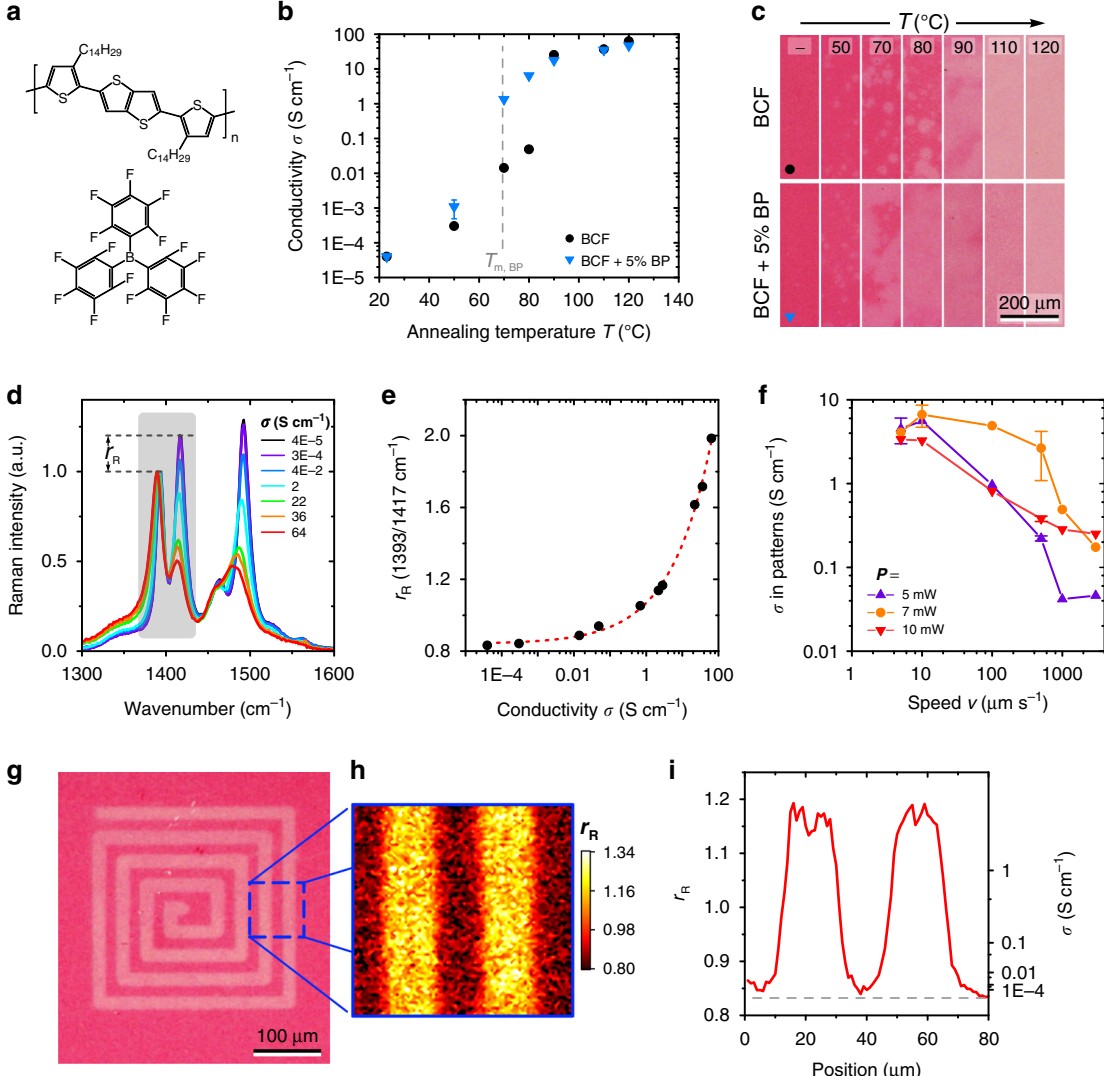

**Fig. 4 Electrical conductivity patterning in PBTTT films by local doping. a** Chemical structures of PBTTT and BCF. **b** Conductivity $\sigma$ of PBTTT films doped through gate by annealing for 1 min at temperature $T$. Solution-deposited dopant layers comprised BCF (black dot) and BCF + 5 wt% biphenyl ('BP'; blue triangle) added as a co-solvent. Dashed line indicates the melting temperature of BP. **c** Transmitted-light micrographs for the films in **b**. **d** Selected Raman spectra of doped PBTTT films, with conductivities indicated. Spectra are normalised by the intensity of the peak at ~1393 cm$^{-1}$; shaded region highlights the peaks used for estimating conductivity. **e** Raman intensity ratio $r_R$ of the maximum intensities of the ~1393 cm$^{-1}$ and ~1417 cm$^{-1}$ peaks of PBTTT as a function of $\sigma$. Dotted line indicates a curve fit to the data. **f** Maximum conductivities obtained in line patterns of doping induced by scanning laser excitation at 532 nm for the indicated incident power $P$ and writing speed $v$. Conductivities were extracted by Raman mapping of $r_R$ using the calibration data in **e**. **g** Transmitted-light micrograph of laser-patterned ($P = 7$ mW, $v = 3$ μm s$^{-1}$) RFID-antenna-type structure. **h** Raman map of $r_R$ for the indicated area and (**i**) the corresponding average profiles of $r_R$ (left ordinate) and $\sigma$ (right ordinate).

solutions in polar solvents. On the other hand, conventional deposition directly onto the hydrophobic PBTTT surface yields strongly dewetted BCF overlayers that are of little practical use (Supplementary Fig. 11).

Overall, the demonstrated molecular-gate-based approach to patterning electrical conductivity (and, more generally, material composition) opens numerous avenues for applications, being simpler than the established methods which typically rely on non-trivial photochemistry or require post-patterning back-filling with pristine material[4,49–51]. PBTTT films doped via molecular gate exhibit excellent long-term stability, with their electrical conductivity remaining essentially unchanged after aging for six months (Supplementary Fig. 12). More generally, in the absence of specific host-donor interactions as in the present example, the thermal stability of material composition patterns would be governed by the glass transition temperature ($T_g$) of the

polymer[52] and the resulting $T_g$ of the polymer-donor blend[53], which can be optimised for specific applications via material selection.

Further optimisation of the process is possible, as evidenced by the highest laser-patterned conductivities falling short of those recorded for macroscopically doped samples (cf. Fig. 4b, f). A promising approach may involve the addition of a small amount of a solid 'co-solvent' to BCF (+5 wt% biphenyl (BP)), which, as shown in Fig. 4b, c and Supplementary Fig. 13, lowers the annealing temperature and time required for reaching a given conductivity by enhancing diffusion of BCF through the gate.

**Beyond 'binary' patterning**. The above-demonstrated use of multiple small-molecular compounds (i.e., BCF + BP) comprising the donor layer can be extended to provide on-demand patterning of several types of functionalities in a single processing

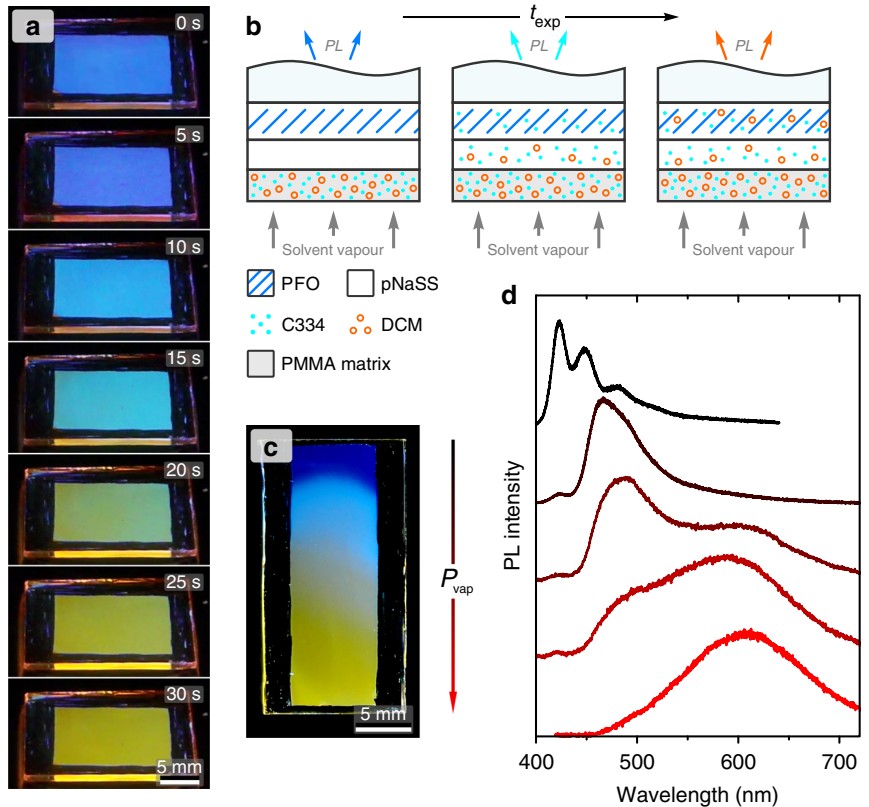

**Fig. 5 One-step broad PL emission tuning in PFO-based blend films. a** Photographs of an inverted PFO/pNaSS/dye-blend trilayer on a glass substrate taken under UV illumination during exposure to 1:1 acetone : water vapour at the indicated times. **b** Schematic illustration of the process in **a**, showing the solvent-vapour-induced (quasi-)sequential diffusion of the cyan- and orange-emitting dyes with increasing exposure time, $t_{exp}$. **c** Photograph of a similarly processed PFO-based blend film but with relative solvent vapour pressure, $P_{vap}$, increasing from top to bottom of the sample, resulting in a gradient of PL emission colours. **d** PL spectra recorded for the sample in **c** spanning positions from top to bottom.

step—i.e., without the need for time-consuming post-processing or re-alignment. This approach relies on identifying small-molecular compounds with desired functionalities, and said compounds possessing sufficiently different physico-chemical properties to enable their (quasi-)discrete, time-separated diffusion through the gate.

Here we demonstrate this concept by broad tuning of PL emission from an active layer via control of its ternary composition. It is well known that adding small amounts of a green-emitting dye to a blue emitter with cascading energy levels can lead to energy transfer from the blue to the green emitter and therewith green-dominated emission. This can be extended to red emission with an appropriate choice of dyes[54]. As an example, we show broad PL tuning in PFO-based blend films. Here the donor layer deposited onto the molecular gate comprises a 2 : 1 wt/wt blend of Coumarin 334 ('C334'; cyan-emitting dye) and 4-(dicyanomethylene)-2-methyl-6-(4-dimethylaminostyryl)-4*H*-pyran (DCM) (orange-emitting dye) (chemical structures and reference PL spectra are shown in Supplementary Fig. 14). The donor layer additionally contains mandelic acid (solid co-solvent) and poly(methyl methacrylate) (PMMA), with the latter added to prevent dye-blend dewetting (Supplementary Fig. 15). Diffusion of the dyes into PFO is then enabled by exposure of the trilayer structure to a vapour of acetone: water (as in Fig. 1b). Of the selected dyes, C334 has a somewhat smaller molar volume than DCM as inferred from their molar masses (283.3 and 303.4 g mol$^{-1}$ respectively), as well as higher relative solubility in acetone due to the presence of carbonyl groups in its chemical structure. Hence, C334 can be expected to diffuse substantially faster than DCM via the gate into the PFO layer,

satisfying the requirements for (quasi-)discrete patterning of the two dyes.

Images of the trilayer structure under UV light illumination taken in real time during continuous exposure to solvent vapour are shown in Fig. 5a, with the process illustrated schematically in Fig. 5b (see also Supplementary Movie 3). Initially (time $t = 0$–5 s), the sample exhibits the characteristic blue PL of PFO. It is noteworthy that at this stage the PL of dye-blend layer is not apparent due to the fact that the images are recorded for the inverted sample (with PFO facing up), as well as concentration-induced quenching of dye PL. Subsequently the PL of the sample turns cyan ($t = 10$–15 s) due to diffusion of C334 in PFO and the resulting PFO→C334 energy transfer occurring for excitation at 365 nm. Finally ($t = 25$–30 s), the PL of the sample turns orange, signifying diffusion of DCM into the PFO layer, with the emission governed by PFO→(C334→)DCM energy transfer.

As a patterning example for the approach presented above, Fig. 5c shows an image of a PFO-based blend film (following removal of the dye-blend donor layer) under UV light illumination that was patterned by solvent vapour exposure through a slot-die placed at the lower end of the substrate. In this configuration, the relative solvent vapour pressure $P_{vap}$ (i.e., ratio of vapour pressure at a given location on the sample to that of the saturated stream) varies across the sample, producing a gradient of PL emission colour. Locally acquired PL spectra (Fig. 5d) exhibit position-dependent variation of the relative amplitudes of individual spectral contributions, with progressive quenching of PFO and C334 emission with increasing $P_{vap}$ due to excitation energy transfer to the energetically lower-lying species.

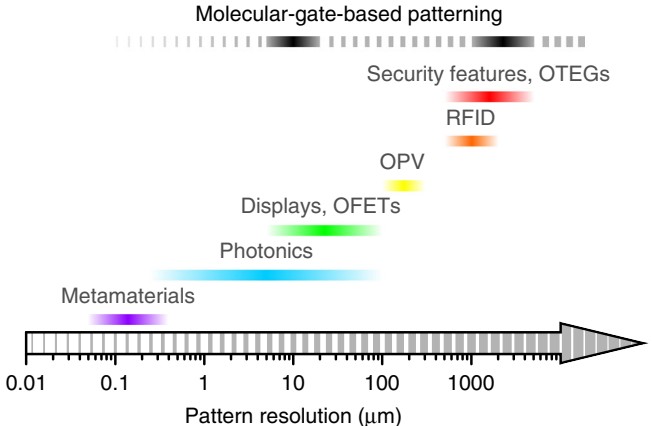

**Fig. 6 Typical length scales required for patterning organic semiconductors.** The characteristic dimensions are shown for metamaterials, photonic structures, displays, organic field-effect transistors (OFETs), organic photovoltaics (OPV), RFID tags, security features and organic thermoelectric generators (OTEGs). Pattern resolutions obtained in the present work are indicated by solid black bars, with broken bars showing their plausible extensions.

**Comparison with state-of-the-art methods**. Conceptually, the closest analogue to the described method in the state-of-the-art is LIFT[17] and its variants such as laser-induced thermal imaging (LITI)[18,55] in which optical heating induces material transfer between a donor sheet and a target substrate placed in close proximity or direct contact. The distinguishing features of molecular-gate-based patterning with respect to LIFT and LITI include lower laser powers, avoiding material degradation inherent to laser-induced donor vaporisation in LITI, molecule-by-molecule rather than dropwise material transfer that offers finer process tuneability, and the capability of patterning both microstructure and material composition. A further difference with most state-of-the-art techniques—which involve physical (additive or subtractive) patterning to produce bilayer-type structures—is that the demonstrated patterning method retains film planarity by donor incorporation within a target film. While retaining the planar film format is advantageous for straightforward integration into multilayer structures, its drawbacks may include issues such as cross-talk which, for instance in the case of OLED displays, requires physical separation between adjacent pixels. One may envisage, however, that leakage can be minimised when the method is combined with laser ablation or for particular geometries given substantial contrast between patterned and baseline regions in terms of, e.g., electrical conductivity (doping patterns) or charge-carrier mobility (chain orientation patterns).

In terms of throughput, for instance in the context of OLEDs, typical material deposition speeds by inkjet printing, slot-die coating and gravure printing are in the $0.1–5\ \mathrm{m\ s^{-1}}$ range[23]. Evidently, the speeds used in this first demonstration of molecular-gate-based patterning ($\leq 3\ \mathrm{mm\ s^{-1}}$) fall short of those for these established techniques. Further optimisation of laser excitation by using high-power sources split into multiple, individually addressable spots as employed in LITI[18] may bring the throughput of the method to the moderate values of $\sim 0.1\ \mathrm{m^2\ min^{-1}}$ associated with laser-based thermal transfer methods[6], while even higher throughputs are common in other state-of-the-art laser structuring methods[56,57]. By comparison, typical throughput values for inkjet printing[2] are $\sim 30\ \mathrm{m^2\ min^{-1}}$. Nevertheless, the use of solution-based deposition and removal of auxiliary layers implies that molecular-gate-based patterning can be configured as a serial process, rather than a batch-type LITI

process, and therefore be compatible with roll-to-roll fabrication[58]. The use of co-solvents (as demonstrated for the case of doping; see Fig. 4b and Supplementary Fig. 13) may be further optimised to improve throughput.

**Applications**. The field of molecular electronics and photonics offers numerous potential applications for the demonstrated method: from the design of photonic structures via chain-conformation patterning[8,33] and displays[59,60] to the fabrication of high-performance field-effect transistors and organic thermoelectric generators by harnessing the capability for local doping[61,62], as well as manipulating electronic[11,63] and thermal[13] conductivities via molecular orientation. Figure 6 shows typical feature sizes required for a selection of electronic and photonic devices and structures. With patterning on $4–25\ \mu m$ and millimetre scales demonstrated herein, the molecular-gate-based approach appears to cover most of the range. Extensions to intermediate and larger scales can be plausibly enabled by modification of stimulus to de-focussed laser beams, solvent vapour jets and heated stamps. Employing deep UV excitation, as in state-of-the-art photolithography, as well as using immersion objectives can enable resolutions $<1\ \mu m$. Stimuli such as electron beams and heated AFM tips may enable feature sizes down to the thickness of the molecular gate itself ($\sim 100\ nm$), albeit with inevitably compromised throughput.

Given the generality of the underlying principles, applications can extend to other fields. The small-molecular donor layer may comprise compounds such as catalysts, cross-linkers or, in fact, reactive monomers for subsequent polymerisation into functional superstructures[64] within the target film. Similarly, the target film can take form of a wide range of (macro-)molecular materials. Hence, one may conceive applications such as patterning polarised security features[29,65] for banknotes and personal ID documents or patterning of antigens in immobilised layers of antibodies for biosensing[66]. Elsewhere, ambient-responsive dyes patterned in commodity polymers can find applications in smart packaging[67].

## Discussion

In summary, we have demonstrated a versatile method for spatial patterning of microstructure and material composition in organic semiconductors by employing a solution-processed molecular-gate interlayer. Patterning of all principal feature types was demonstrated, including chain conformation and orientation, and material composition. Heat-, solvent-vapour- and laser-based implementations were presented, with minimum resolution $<5\ \mu m$ and writing speeds up to $3\ \mathrm{mm\ s^{-1}}$. A number of further advantages of the molecular-gate-based approach were highlighted, such as the capacity for one-step (quasi-)discrete patterning of multiple functionalities or components. The demonstrated method is expected to provide a practical means to micro-patterning of organic semiconductors in both roll-to-roll and laboratory-scale environments for applications in the field of molecular electronics and beyond.

## Methods

**Materials**. pNaSS (MW $\sim 1 \times 10^6\ \mathrm{g\ mol^{-1}}$), PFO, P3HT (regioregular), PMMA (MW $\sim 15{,}000\ \mathrm{g\ mol^{-1}}$) and C334 were purchased from Aldrich. LA (>98%), BT (>99%), BCF (>98%), BP (>99.5%), DCM and ʟ-(+)-mandelic acid (>99%) were purchased from TCI Chemicals. PBTTT was purchased from 1-Material. ITO/glass substrates (ITO thickness $\sim 100\ nm$) were purchased from Ossila. All materials were used as received. Details of the employed solvents are given in the Supplementary Note 1.

**Sample fabrication**. pNaSS 'molecular-gate' films were spin-coated from 2 to 4 wt% solutions in deionised water with Triton X-100 surfactant added at $\sim 1$ wt% relative to the amount of solvent. pNaSS solutions were filtered (polyethersulfone (PES),

0.45 μm pore size) prior to use. For all functional small-molecular donor layers, spin-coating of the respective solutions was carried out by dynamic deposition—i.e., onto the substrate at the target spin speed. Laser patterning was carried out using a WITec Alpha 300RA instrument using its stepper-motor-driven stage for scanning the sample in the laser focal plane. Continuous-wave (CW) laser excitation was used throughout.

**Patterning chain conformation**. PFO films were spin-coated from 2 wt% solutions in toluene at 2000 r.p.m., with both solution and substrates pre-heated to 70 °C, resulting in ~85 nm-thick glassy films. Glass and ITO/glass substrates were used for macroscopic and local chain-conformation switching, respectively. pNaSS films were spin-coated from 4 wt% solutions at 3000 r.p.m. (thickness ~200 nm). LA donor layers were spin-coated from 20 wt% solutions in acetone at ~8000 r.p.m. Diffusion of LA into PFO and, therewith, formation of β-phase was accomplished macroscopically by heating to 50 °C. Local formation of β-phase was enabled by exposure to laser excitation (785 nm; power $P = 10–50$ mW) focussed through the substrate onto the PFO film and scanned at speeds $v = 0.1–10$ μm s$^{-1}$ and, elsewhere, by nozzle-based exposure to solvent vapour of a 1 : 1 vol/vol acetone : water solution through which N$_2$ carrier gas was bubbled. Note that, in the absence of LA, neither acetone nor water in liquid or vapour form would induce β-phase formation, being non-solvents for PFO[34]. In all cases, LA and pNaSS were removed after patterning by spin-off with, sequentially, acetone, water and—again—acetone.

**Patterning chain orientation**. Proof-of-principle P3HT : BT blend films were spin-coated from 2.5 wt% solutions in 2 : 1 wt/wt BT : ethyl acetate at 2000 r.p.m., with solution and substrates held on hotplates at 120 °C and 40 °C, respectively, prior to deposition. P3HT films were spin-coated from 2 wt% solutions in chlorobenzene at 4000 r.p.m., with both solution and ITO/glass substrates pre-heated to 50 °C, yielding ~90 nm-thick films. pNaSS films were spin-coated from 4 wt% solutions at 7000 r.p.m. (thickness ~130 nm). BT donor layers were deposited by casting ~20 μL of 40 wt% solutions in methylene chloride (with 1% of Triton X-100) onto films spinning at ~700 r.p.m. and, following solvent evaporation over ~60 s, placing a 170 μm-thick glass coverslip onto the still-liquid BT layer and initiating its crystallisation. The function of the coverslip was to prevent excessively rapid sublimation of BT during subsequent processing. Laser patterning of chain orientation was performed using excitation at 785 nm ($P = 55$ mW) focussed through the substrate onto the P3HT film and scanned at $v = 0.1–3$ mm s$^{-1}$. After patterning, the coverslip was peeled off and, following rapid sublimation of BT under ambient conditions, pNaSS gate was removed by spin-off with water.

**Patterning electrical conductivity**. PBTTT films were spin-coated from 1.5 wt% solutions in chlorobenzene at 4000 r.p.m., with both solution and glass substrates pre-heated to 110 °C, resulting in ~43 nm-thick films. The films were subsequently crystallised by annealing at 180 °C for 30 min under N$_2$ atmosphere followed by slow cooling to room temperature. pNaSS films were spin-coated from 2 wt% solutions at ~6000 r.p.m. (thickness ~80 nm). BCF donor layer were spin-coated from 60 wt% solutions in 10 : 1 wt/wt diethyl ether : methanol at 3000 r.p.m. (doping by thermal annealing) or 9000 r.p.m. (laser-based doping). Elsewhere, 5 wt% of BP (relative to the amount of solvent) was added to BCF solutions to act as a co-solvent at processing temperatures above its melting temperature ($T_m = 69$ °C). Macroscopic doping was enabled by heating the trilayer samples in air to selected temperatures (50–120 °C) for 1 min. Laser patterning of doping was performed using excitation at 532 nm ($P = 5–12.5$ mW) focussed 20 μm above the sample surface (×40 objective; NA = 0.6) to yield larger feature size for improved analysis and visualisation, and scanned at $v$ spanning 5 μm s$^{-1}$ to 3 mm s$^{-1}$. After patterning, BCF and pNaSS were removed by spin-off with, sequentially, acetonitrile and water.

**Patterning PL emission in PFO-based blends**. PFO films were deposited on glass substrates as described above. pNaSS films were spin-coated from 4 wt% solutions at 3000 r.p.m. (thickness ~200 nm). The dye-blend solution comprised 2 wt% C334 (cyan-emitting dye) in 1 : 2 wt/wt acetone : methylene chloride with 1 wt% DCM (orange-emitting dye), 1 wt% mandelic acid (co-solvent; $T_m = 135$ °C) and 4 wt% PMMA (matrix polymer) added relative to the amount of solvent. The dye-blend solution was spin-coated at 6000 r.p.m. (Quasi-)discrete diffusion of the C334 and DCM dyes into the PFO film was achieved by exposure to vapour of 1 : 1 acetone : water (as above) directed at the sample via a large-opening tube (homogeneous PL modification) or a slot-die (graded PL modification). For the latter example, spin-off of the residual dye-blend donor layer was performed with methylene chloride; pNaSS was not removed to avoid further, spin-off-induced changes in PL due to the non-negligible solubility of C334 in water. Reference C334 and DCM films were spin-coated at 1500 r.p.m. from 0.1 wt% solutions in 1 : 2 wt/wt acetone : methylene chloride that additionally contained 5 wt% PMMA (relative to the amount of solvent) to maintain dilution of the dyes in the solid state.

**Characterisation**. Unless otherwise noted, all characterisation of semiconducting polymer films was performed after removal of any auxiliary layers (i.e., molecular gate and donor layer) by spin-off. Whenever shown, error bars indicate standard deviation (SD) from analysis on different areas of a given sample. WITec Alpha

300RA instrument (reflection geometry, scanning mode) was used for spectroscopic PL ($\lambda_{ex} = 355$ nm) and Raman mapping analyses (CW excitation in all cases); additional PL spectra were recorded under UV light illumination ($\lambda_{ex} \approx 365$ nm). The excitation wavelengths for Raman analysis were as follows: 488 nm (PBTTT), 633 nm (PFO) and 785 nm (P3HT). Polarised Raman analysis was performed with synchronous adjustment of both excitation and detection polarisation. Incident laser power and integration time were optimised to avoid sample degradation by, e.g., de-doping. Small-area transmitted-light microscopy was performed using a BX51 instrument (Olympus); large-area reflected-/transmitted-light microscopy was performed with a Mantis Elite stereo-microscope (Vison Engineering). Vis-NIR transmission spectra were recorded with a Bruker Vertex 70 FTIR spectrophotometer coupled to a Bruker Hyperion optical microscope. Film thickness was measured using a DektakXT profilometer (Bruker). AFM was performed in tapping mode with a Keysight 5500 instrument (Agilent). Electrical conductivity of doped PBTTT samples was measured with an Ecopia HMS-5000 instrument using the four-probe van der Pauw method for square ~6 × 6 mm$^2$ film areas with silver paste contacts placed on each corner. For neat, undoped PBTTT—the conductivity of which could not be measured directly—the literature value[43] of $\sigma = 4 \times 10^{-5}$ S cm$^{-1}$ was assumed. Differential scanning calorimetry (DSC) was performed using a Mettler-Toledo DSC 2 instrument calibrated using indium standards. BT and P3HT were loaded into low-pressure aluminium pans and sealed to prevent evaporation of BT at elevated temperatures (confirmed by weighing the pans before and after measurement). Samples were cycled in the −30 to 180 °C range (−30 to 260 °C for neat P3HT in pierced crucibles under N$_2$ flow) at 10 °C min$^{-1}$.

## Data availability

All data supporting the findings of this study are available within the article and its Supplementary Information, or from the corresponding authors upon reasonable request.

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

## Acknowledgements

We are indebted to Enrique Pascual (ICMAB-CSIC) for providing insights into laser patterning and Lou Kiss (Middlesex University) for freely shared advice on the preparation of the graphics. We express our deepest gratitude to Paul Smith (ETH Zürich) for critically reviewing the manuscript. We thank Miquel Casademont-Viñas (ICMAB-CSIC) for invaluable contribution to exploring the limits of the method, and Agustín Mihi and Andrés Gómez (ICMAB-CSIC) for assistance with optical spectroscopy and AFM analysis. We also thank Daniel Kremer and Hans-Werner Schmidt (Universität Bayreuth) for their customary hospitality and assistance with DSC measurements. We further acknowledge financial support from the Spanish Ministry of Economy, Industry and Competitiveness through the "Severo Ochoa" Programme for Centers of Excellence in R&D (SEV-2015-0496) and project reference PGC2018-095411-B-I00, as well as the European Research Council (ERC) under grant agreement number 648901.

## Author contributions

A.P. and M.C.Q. conceived the project and designed the experiments. A.P. fabricated the samples, performed the experiments and analysed the data. A.P. and M.C.Q. wrote the manuscript.

## Competing interests

A patent application (EP20382514) has been filed based on the results reported herein. The authors declare no competing interests.
