## [Peer Review File · Nature Communications]

Reviewers' Comments:

Reviewer #1:

Remarks to the Author:

The authors reported organic molecular patterning concept using solution-processed 'molecular gate' interlayer between target layer and donor layer. The molecular gate serves as a semipermeable membrane which prevents uncontrolled diffusion of the functional compound in donor layer into the target layer, and only passes the diffusion of functional compound when suitable stimulus is applied, which is laser induced heat in this study. The demonstrated data show that this concept can change the local properties of target layer, such as crystallization, chain orientation, and doping.

Although this manuscript proposes interesting approaches and results about organic material layer patterning using molecular gate concepts, the quality of this manuscript seems not suitable for publication in Nature Communications due to the following issues.

1. The most critical issue is about the title and main storyline of manuscript. Authors emphasize that this method can combine the spatial resolution of photolithography and versatility of printing method. However, this method is fundamentally different from photolithography and printing method, therefore, that kind of comparison cannot be made reasonably. For the fabrication of various electronic devices, photolithography and printing are usually utilized to pattern target layer 'physically' for the formation of desired shapes or structures, and this is one of the most important steps in the fabrication processes. For example, semiconducting layer of each unit channel in FET arrays or LED arrays should be physically separated from each other. However, the method proposed in this manuscript is just about changing local properties or components of target layer using selective diffusion of donor compound which can be considered as chemical patterning rather than physical patterning of target layer. For more certain comparison, this method can pattern the P3HT layer with selective dopant diffusion to have partially different conductivity area, however, P3HT layer itself cannot be patterned, while printing can make it possible. Due to this issue, current title and the direction of main storytelling which includes comparison between photolithography and printing should be revised to emphasize the novelty of chemical patterning.

2. Due to the above issue, this method cannot have any novelty when considering conventional photolithography, printing, and LIFT methods which are usually used for fabrication of electronic devices.

3. In the manuscript, it is written that small functional molecules of donor compounds are diffused through molecular gate by photo-induced thermal and solvent-vapor-based stimuli. However, in my opinion, it seems quite difficult to control pinpoint diffusion because heat and solvent-vapor spread easily and cannot be controlled minutely, which may cause inevitable unintended diffusion of small molecules into target layer. Although the authors proposed the power and moving speed of laser are the key parameter to control the diffusion, other parameters which may affect small molecule diffusion properties or techniques to prevent spread of heat and solvent-vapor should be proposed for fine and high-resolution patterning.

4. In addition, the stability issue of the patterned layer should also be addressed. It seems that small molecules diffused into the target layer can be diffused to the non-patterned area of target layer after patterning process due to the external stimuli, such as heat, or the elapse of time. If diffused small molecules can spread into the other area of target layer after patterning, this method cannot be utilized. Therefore, I recommend authors add the stability data of the patterned layer.

In my opinion, due to the above issues, this manuscript is not suitable for publication in Nature Communications.

Reviewer #2:

Remarks to the Author:

Perevendentsev et al discuss a technique for the patterning of organic semiconductors. They propose a sequential solution-based deposition on a target semiconductor layer using an interlayer followed by a functional layer. Some points are noted below:

1. While this appears to be a promising technique, the presentation of the basic concept and its application are somewhat difficult to follow. The authors have developed a strategy using terminology that is not easy to understand. The schematic shown in Figure 1 should better help in demonstrating the technique, perhaps with simple or complex features formed. With a scale bar of 5 mm, it is unclear how this is a micropatterning approach. Importantly, it would be very helpful if the patterning demonstration of this technique could be clearer (other than Figure 4g, the claim of high-resolution patterning is not borne out).
2. The authors discuss the potential issue of delamination of adjacent layers upon physical aging or flexure. However, in this work, they have merely speculated on the stability of their technique (e.g. "incorporation of donor compounds within the target film"). No experimental data has been provided to support their conjecture. Given that this is an important concern for such systems, the stability to either flexure or aging or both needs to be demonstrated.
3. Can this technique be used on substrates other than ITO/glass as demonstrated? For translation to "roll-to-roll", this discussion would be pertinent.
4. Figure 6 is confusing. How were the "plausible extensions" determined? It is unclear how the technique is extensible to metamaterials for instance.
5. What is the height of the features that can be formed?
6. There is extensive discussion on the speed of this technique. However, in the absence of context or comparison to other methods, the advantage is not clear. By switching to electron beam or AFM, this will likely change (p. 23).
7. There are also several works in the area of patterning various forms of organic electronics, including but not limited to conducting polymers. The authors should consider these to establish a clear context for this technique (ref 9 from Menard is a good start but is dated – several papers since then have advanced the field). This holds for many of the references cited which are from the previous decade. Given the sophistication of some of these reports, the advance of this work needs to be more clearly demonstrated.

Reviewer #3:

Remarks to the Author:

In their manuscript the authors describe a process driven by laser light or solvent vapor jets to achieve local diffusion of functional small-molecular compounds through a solution-processed 'molecular gate' interlayer. The authors show how this diffusion process can be used to pattern organic semiconductor thin films. The process can also be used for the diffusion of a functional donor compound by locally applying heat or solvent vapor to modify the conformation, orientation or crystallization of the polymer chains in the target films. Furthermore, local doping of the target films is also demonstrated with this process. The process described is novel, original in its implementation and clearly of interest to the organic electronic community.

To strengthen their discussion the authors could consider providing more details on patterning speeds vs. minimum feature size and since this technique is a serial patterning process how its write speed compares to parallel patterning processes such as photo-lithography. Since the process seems to be diffusion driven, then more details on the patterning speed in terms of diffusion lengths (and temperature) would be useful as well. Presenting these different parameters on a table would help to better understand the capabilities of the process and how it compares to other serial patterning or direct-write techniques.

Referee #1:

The authors reported organic molecular patterning concept using solution-processed 'molecular gate' interlayer between target layer and donor layer. The molecular gate serves as a semipermeable membrane which prevents uncontrolled diffusion of the functional compound in donor layer into the target layer, and only passes the diffusion of functional compound when suitable stimulus is applied, which is laser induced heat in this study. The demonstrated data show that this concept can change the local properties of target layer, such as crystallization, chain orientation, and doping.

Although this manuscript proposes interesting approaches and results about organic material layer patterning using molecular gate concepts, the quality of this manuscript seems not suitable for publication in Nature Communications due to the following issues.

1. The most critical issue is about the title and main storyline of manuscript. Authors emphasize that this method can combine the spatial resolution of photolithography and versatility of printing method. However, this method is fundamentally different from photolithography and printing method, therefore, that kind of comparison cannot be made reasonably. For the fabrication of various electronic devices, photolithography and printing are usually utilized to pattern target layer 'physically' for the formation of desired shapes or structures, and this is one of the most important steps in the fabrication processes. For example, semiconducting layer of each unit channel in FET arrays or LED arrays should be physically separated from each other. However, the method proposed in this manuscript is just about changing local properties or components of target layer using selective diffusion of donor compound which can be considered as chemical patterning rather than physical patterning of target layer. For more certain comparison, this method can pattern the P3HT layer with selective dopant diffusion to have partially different conductivity area, however, P3HT layer itself cannot be patterned, while printing can make it possible. Due to this issue, current title and the direction of main storytelling which includes comparison between photolithography and printing should be revised to emphasize the novelty of chemical patterning.

We thank the referee for suggesting this re-write, and believe that the focus of the revised manuscript is substantially clearer. We have changed the title and extensively re-written the Introduction and Discussion sections to remove direct parallels drawn between the demonstrated method with photolithography and printing. Comparisons with photolithography and printing were retained only to benchmark the performance of the method against these established techniques.

We have emphasized that the method patterns physical properties within a planar film format unlike the conventional additive/subtractive bottom-up and top-down fabrication methods, highlighting the practical advantages and disadvantages this may offer. The pertinent example provided by the referee on cross-talk issues in OLEDs has been included as part of Discussion. Nevertheless, we emphasize that cross-talk can be minimized for specific geometries given substantial (e.g. more than an order of magnitude) contrast between patterned and baseline semiconductor regions in terms of, for example, electrical conductivity (doping patterns; Fig. 4) or charge-carrier mobility (chain orientation patterns; Fig. 3). Furthermore, certain device features such as waveguides (via chain conformation patterning; Figs. 1–2) would, in fact, benefit from retained planarity by avoiding the necessity for back-filling or the use of reactive-ion etching which can lead to severe device degradation.

The referee is only partly correct in describing the demonstrated method as chemical patterning. In the revised version of the manuscript we have clarified this point, explaining that via physico-chemical interaction between a given donor and an organic semiconductor we are, in fact patterning on-demand *physical* (e.g. chain orientation, chain conformation or the degree of crystallinity) or *chemical* (e.g. doping) properties. We have clarified that the specific type of patterning is accomplished by selection of an appropriated donor compound and whether or not it is removed post-patterning by, e.g., sublimation or exposure to a given solvent.

2. Due to the above issue, this method cannot have any novelty when considering conventional photolithography, printing, and LIFT methods which are usually used for fabrication of electronic devices.

The referee provides limited examples of specifically top-down or bottom-up fabrication methods which generally pattern material composition. We have emphasized in the revised text that one of the novel aspects of the method is the capability for on-demand patterning microstructure or material composition – or, in fact, both simultaneously. Moreover, we have commented on the substantial differences with LIFT/LITI methods mentioned by the referee, including that the demonstrated method (i) necessitates the use of lower laser powers (since donor vaporisation is not required) which avoids photochemical degradation of donor and (ii) features diffusion-mediated molecule-by-molecular (rather than dropwise) material transfer which enables a finer control of the process down to, essentially, molecule-on-demand precision.

3. In the manuscript, it is written that small functional molecules of donor compounds are diffused through molecular gate by photo-induced thermal and solvent-vapor-based stimuli. However, in my opinion, it seems quite difficult to control pinpoint diffusion because heat and solvent-vapor spread easily and cannot be controlled minutely, which may cause inevitable unintended diffusion of small molecules into target layer. Although the authors proposed the power and moving speed of laser are the key parameter to control the diffusion, other parameters which may affect small molecule diffusion properties or techniques to prevent spread of heat and solvent-vapor should be proposed for fine and high-resolution patterning.

We note that even in the absence of full optimization in terms of, e.g., laser parameters and focussing, we have conclusively demonstrated pattern resolutions down to $\sim 4 \mu\text{m}$ (see Fig. 2b,c or Fig. 3f) – that is, approaching the diffraction limited resolution of $\sim 1 \mu\text{m}$ diameter (see also response to reviewer #2). Contrary to the referee's comment, this indicates that donor diffusion can be indeed controlled minutely. This is understood to be due to the Arrhenius-type exponential dependence of diffusion coefficient on temperature, which results in drastically slowed-down diffusion rates of donor outside, for instance, a laser-illuminated area. In the case of solvent vapour, parasitic lateral diffusion of solvent molecules can be minimized, for example, by using a confining sheath gas. The latter is commonly employed in, e.g., aerosol-jet printing to enable pattern resolutions down to $10 \mu\text{m}$, a reference for which is provided in the revised text.

Furthermore, in the revised manuscript we point out in the example shown in Fig. 1a, panel 5, that PL emission colour is very homogeneous for the target film area where donor diffusion occurred through the gate, signifying a highly controllable process. On the contrary, PL emission colour is visibly less homogeneous for the sample area where donor was deposited directly onto the target film.

We have emphasized the points above in the revised version of the manuscript. Furthermore, we have added new data for the exemplary case of laser-patterning doping in PBTTT (Supplementary Figure 9 and Supplementary Table 1) showing that the lateral, in-plane diffusion component does not significantly impact on pattern resolution for the case of light-to-heat conversion using resonant excitation.

Further exploration of diffusion phenomena in these multilayer structures certainly presents an interesting direction for future work which, however, is beyond the scope of the present manuscript.

4. In addition, the stability issue of the patterned layer should also be addressed. It seems that small molecules diffused into the target layer can be diffused to the non-patterned area of target layer after patterning process due to the external stimuli, such as heat, or the elapse of time. If diffused small molecules can spread into the other area of target layer after patterning, this method cannot be utilized. Therefore, I recommend authors add the stability data of the patterned layer.

At the referee's request we have added exemplary data for long-term stability of electrical characteristics of PBTTT films doped via molecular gate (Figure 11 in the revised Supplementary Information). The data shows that electrical conductivity remains unchanged within a few percent after aging the samples in ambient atmosphere over a period of up to six months.

We have also expanded the discussion in the text, including additional references, commenting that—in the absence of specific donor-host interactions as in the case of BCF-doped PBTTT—the stability of material composition patterns is governed by the glass transition temperature, T_g , of the polymer and, hence, that of the blend. This is well-documented in the literature for polymer:fullerene-based bulk heterojunction blends which are known to be stable against coarsening of fullerene domains provided that they are not heated above T_g of the polymer – references to which are provided in the revised text.

Finally, we note that the referee's comment only applies for the specific case of patterning material composition by the present method. For the examples showing the patterning of microstructure (Figures 1–3) the small-molecular donor is subsequently removed from the films and the physical patterning obtained (e.g. orientation and conformation) is thermodynamically stable.

Referee #2:

Pereventsev et al discuss a technique for the patterning of organic semiconductors. They propose a sequential solution-based deposition on a target semiconductor layer using an interlayer followed by a functional layer. Some points are noted below:

1. While this appears to be a promising technique, the presentation of the basic concept and its application are somewhat difficult to follow. The authors have developed a strategy using terminology that is not easy to understand. The schematic shown in Figure 1 should better help in demonstrating the technique, perhaps with simple or complex features formed. With a scale bar of 5 mm, it is unclear how this is a micropatterning approach. Importantly, it would be very helpful if the patterning demonstration of this technique could be clearer (other than Figure 4g, the claim of high-resolution patterning is not borne out).

We have updated the schematics to illustrate the method more clearly; these are found in Fig. 1a,b and Fig. 2a. Captions and accompanying text were modified to further clarify the concept and terminology, with the latter made consistent throughout the text as well as with common use in the field.

We have added an animated depiction of the method and its applications (Supplementary Video 1) as well as an additional step-by-step schematic illustration for the case of laser-based patterning to Supplementary Fig. 1.

We have also clarified in the text that the first proof-of-concept example in Fig. 1 is intentionally *macroscopic*. This example is intended to be maximally visual with the aim of, at this point in the manuscript, avoiding a distracting lengthy interpretation of micro-scale imaging data.

Later in the manuscript, high-resolution patterning is demonstrated for all primary feature types:

- chain conformation (Fig. 2b,c – minimum resolution $\approx 4 \mu\text{m}$);
- chain orientation (Fig. 3f – minimum resolution $\approx 5 \mu\text{m}$), and
- doping (Fig. 4g – minimum resolution $\approx 18 \mu\text{m}$; Supplementary Fig. 9b – minimum resolution $\approx 7 \mu\text{m}$).

2. The authors discuss the potential issue of delamination of adjacent layers upon physical aging or flexure. However, in this work, they have merely speculated on the stability of their technique (e.g. “incorporation of donor compounds within the target film”). No experimental data has been provided to support their conjecture. Given that this is an important concern for such systems, the stability to either flexure or aging or both needs to be demonstrated.

A full study of the mechanical properties of the system goes beyond the scope of the present manuscript. Therefore, to avoid unsubstantiated claims, we have removed the text in question in the revised version of the manuscript.

3. Can this technique be used on substrates other than ITO/glass as demonstrated? For translation to “roll-to-roll”, this discussion would be pertinent.

P-doping in PBTTT (Fig. 4f–i and Supplementary Figs. 8 and 9) was laser-patterned by *resonant* excitation of the polymer wherein the ITO layer was not needed for light-to-heat conversion – PBTTT was deposited on ordinary glass substrates. Furthermore, the example of solvent-vapour-based patterning of ternary blend composition (Fig. 5) also utilizes ordinary glass substrates. Hence flexible plastic substrates such as PET may also be used, provided that the employed processing conditions do not exceed their thermal stability and the employed solvents are selected such that substrate dissolution is avoided.

4. Figure 6 is confusing. How were the “plausible extensions” determined? It is unclear how the technique is extensible to metamaterials for instance.

We have clarified in the text accompanying Fig. 6 that plausible extensions to intermediate/large scales can be enabled via modification of stimulus to, e.g., de-focussed laser beams, solvent vapour jets or heated stamps. Resolutions below 1 μm can be enabled via the use of deep UV laser excitation and immersion objectives, as commonly employed in state-of-the-art photolithography. Resolutions down to approximately the thickness of the molecular gate itself (~ 100 nm) can be enabled via the use of, e.g., heated AFM tips – we have specified that the latter would, of course, inevitably compromise throughput.

5. What is the height of the features that can be formed?

We have emphasized in the revised version of the manuscript that the demonstrated method patterns physical and chemical material characteristics *within a planar film format*. An exemplary analysis of topography for patterned films is shown in Supplementary Fig. 6, confirming the above. To provide thickness steps, one may recall, for instance, the change in solubility enabled by doping, implying that the surrounding area around doping patterns generated with our method could be washed off with common solvent such as chlorobenzene, as was demonstrated recently [Jacobs, I. E. *et al.*, Direct-write optical patterning of P3HT films beyond the diffraction limit. *Adv. Mater.* **29**, 1603221 (2017)].

6. There is extensive discussion on the speed of this technique. However, in the absence of context or comparison to other methods, the advantage is not clear. By switching to electron beam or AFM, this will likely change (p. 23).

In Discussion we have added a comparison of the speed of the method (in its first demonstration presented herein) and its expected throughput with that of established techniques such as inkjet printing and laser-based thermal transfer. We have also emphasized that the use of stimuli such as electron beams or heated AFM tips would inevitably compromise throughput which may be acceptable for certain applications.

7. There are also several works in the area of patterning various forms of organic electronics, including but not limited to conducting polymers. The authors should consider these to establish a clear context for this technique (ref 9 from Menard is a good start but is dated – several papers since then have advanced the field). This holds for many of the references cited which are from the previous decade. Given the sophistication of some of these reports, the advance of this work needs to be more clearly demonstrated.

We understand that when introducing a new technology, benchmarking should be done with respect to already established technologies as well as the state-of-the-art in the academic literature. We have made an effort to make a good compromise between completeness and not making a full review on patterning techniques as the present manuscript is not a review. In this respect, we note that 45 of the 66 references in the submitted manuscript were from the decade of 2010–2020. Nevertheless, in the revised version we have reviewed the cited articles and, where appropriate, replaced several of them with more contemporary reports. Two new up-to-date review articles on patterning techniques for flexible electronics have also been added.

Referee #3:

In their manuscript the authors describe a process driven by laser light or solvent vapor jets to achieve local diffusion of functional small-molecular compounds through a solution-processed ‘molecular gate’ interlayer. The authors show how this diffusion process can be used to pattern organic semiconductor thin films. The process can also be used for the diffusion of a functional donor compound by locally applying heat or solvent vapor to modify the conformation, orientation or crystallization of the polymer chains in the target films. Furthermore, local doping of the target films is also demonstrated with this process. The process described is novel, original in its implementation and clearly of interest to the organic electronic community.

To strengthen their discussion the authors could consider providing more details on patterning speeds vs. minimum feature size and since this technique is a serial patterning process how its write speed compares to parallel patterning processes such as photo-lithography. Since the process seems to be diffusion driven, then more details on the patterning speed in terms of diffusion lengths (and temperature) would be useful as well. Presenting these different parameters on a table would help to better understand the capabilities of the process and how it compares to other serial patterning or direct-write techniques.

We thank the referee for pointing out this important aspect of the demonstrated method, and have added new data (Supplementary Fig. 9 and Supplementary Table 1) to address this issue in more detail. In particular, the data in Supplementary Table 1 for laser-patterning doping in PBTTT shows that the lateral, in-plane diffusion component plays in a negligible role in determining pattern dimensions. For instance, reducing writing speed (i.e. increasing the effective diffusion time) by a factor of 600 leads only to a factor of 1.6 increase in pattern dimensions. The same data shows that laser power plays the primary role in determining pattern dimensions. A full exploration of diffusion characteristics in these multilayer structures will, no doubt, be addressed in the future work.

The revised Discussion section now includes a comparison of the speed of the method (in its first demonstration presented herein) and its expected throughput with that of established techniques such as inkjet printing and laser-based thermal transfer.

Reviewers' Comments:

Reviewer #1:

Remarks to the Author:

In the revised manuscript by Perevedentsev et al., the reviewers' comments and questions have been clearly addressed with additional explanations and experimental data. Therefore, in my opinion, the revised manuscript is suitable for publication.

Reviewer #2:

Remarks to the Author:

The manuscript has been improved. In particular, the schematic illustration of molecular gate concept is a welcome addition given that the text is not easy to follow. There is some question regarding the feature height:

The size of the features is not clear from the text. The authors state "that the demonstrated method patterns physical and chemical material characteristics within a planar film format". However, from the AFM image, it appears that the features are ~2 nm in height. This raises questions on the fidelity of this technique. Given the minimum feature resolution of 4 μm , even 10s to 100s of nm of feature height can form "planar films". Higher thicknesses are speculated on (line 547).

Minor issues:

Introduction -This sentence is questionable "We hold the fact to be self-evident that the field of molecular, 'plastic' electronics and photonics is nearing its promise of delivering several classes of high-performance ..." – A search of the term "plastic electronics" reveals several commercially available products in the marketplace. Indeed, this term is nearly 40 years old. It may be suggested to reword this sentence to better convey the intent here.

Along similar lines, the title "Rapid, high-resolution patterning of microstructure, composition and doping in organic semiconductors using 'molecular gates'" does not seem to be grammatically correct – it seems to imply that the authors are patterning composition and doping. There is a disagreement of subject and predicate.

The prior title - rapid, high-resolution patterning of microstructure and composition in organic semiconductors is less problematic.

Reviewer #3:

Remarks to the Author:

The authors have addressed the reviewer's comments satisfactorily and the revised manuscript and revised supplementary information are now suitable for publication in Nature Communications.

Referee #2:

The manuscript has been improved. In particular, the schematic illustration of molecular gate concept is a welcome addition given that the text is not easy to follow. There is some question regarding the feature height:

The size of the features is not clear from the text. The authors state “that the demonstrated method patterns physical and chemical material characteristics within a planar film format”. However, from the AFM image, it appears that the features are ~2 nm in height. This raises questions on the fidelity of this technique. Given the minimum feature resolution of 4 μm, even 10s to 100s of nm of feature height can form “planar films”. Higher thicknesses are speculated on (line 547).

The referee appears to have misunderstood the AFM data in question and interpreted the topography profile across a surface defect (Supplementary Figure 6c; height ≈ 2 nm) as the patterned feature. We have clarified the manuscript text and the caption of Supplementary Figure 6 to emphasise that AFM is performed across line patterns of chain orientation within (otherwise) isotropic P3HT films. The data in Supplementary Figure 6a,b shows that the total film thickness reduces from ~90 nm for the pristine film to ~88.5 nm at the centre of the line pattern. This small ~2% change of film thickness represents a negligible deviation from film planarity, as written in the text.

Feature resolutions quoted by the referee are *lateral* pattern dimensions (e.g. 4.6 μm for the data in questions), as made clear in the text, and thus are not related to the *vertical* topography profiles.

Minor issues:

Introduction -This sentence is questionable “We hold the fact to be self-evident that the field of molecular, ‘plastic’ electronics and photonics is nearing its promise of delivering several classes of high-performance ...” – A search of the term “plastic electronics” reveals several commercially available products in the marketplace. Indeed, this term is nearly 40 years old. It may be suggested to reword this sentence to better convey the intent here.

We have removed the term ‘plastic electronics’ as it indeed can be considered ambiguous and, arguably, somewhat dated. The first two sentences of the introduction have been modified to reflect the fact that a number of products have recently appeared in the marketplace that can be categorized as “molecular electronics”.

In our modifications we were careful not to overstate the success of the field given that a number of key device types (e.g. organic photovoltaics, wearable electronics and organic thermoelectric power generators) are not yet ubiquitous on the marketplace. Clearly the latter is due to persisting challenges – one of which is the unavailability of suitably versatile micro-fabrication methods, which the present manuscript attempts to address.

Along similar lines, the title “Rapid, high-resolution patterning of microstructure, composition and doping in organic semiconductors using ‘molecular gates’” does not seem to be grammatically correct – it seems to imply that the authors are patterning composition and doping. There is a disagreement of subject and predicate.

The prior title - rapid, high-resolution patterning of microstructure and composition in organic semiconductors is less problematic.

The title has been revised as per referee’s suggestion and to avoid the use of punctuation.